# Observation Time Effects in Reinforcement Learning on Contracts for Difference

Maximilian Wehrmann, Nico Zengeler * and Uwe Handmann

Hochschule Ruhr West, University of Applied Sciences, Duisburger Str. 100, 45479 Mülheim an der Ruhr, Germany; maximilian.wehrmann@stud.hs-ruhrwest.de (M.W.); uwe.handmann@hs-ruhrwest.de (U.H.)
* Correspondence: nico.zengeler@hs-ruhrwest.de; Tel.: +49-208-88-254-816

**Abstract:** In this paper, we present a study on Reinforcement Learning optimization models for automatic trading, in which we focus on the effects of varying the observation time. Our Reinforcement Learning agents feature a Convolutional Neural Network (CNN) together with Long Short-Term Memory (LSTM) and act on the basis of different observation time spans. Each agent tries to maximize trading profit by buying or selling one of a number of contracts in a simulated market environment for Contracts for Difference (CfD), considering correlations between individual assets by architecture. To decide which action to take on a specific contract, an agent develops a policy which relies on an observation of the whole market for a certain period of time. We investigate whether or not there exists an optimal observation sequence length, and conclude that such a value depends on market dynamics.

**Keywords:** machine learning; contracts for difference; deep neural networks





## 1. Introduction

We refer to automatic trading as the ability of a trading system to act based only on given parameters, without any human interference (Venkataraman 2001). Automated trading agents may use a Reinforcement Learning approach with recurrent neural networks to optimize trading strategies (Zengeler and Handmann 2020). We use the notion of Reinforcement Learning, as it allows us to model effects of an agent's action in a simulated environment instead of simply predicting prices. Yet, for our experiments, we assume the effect that single trading actions have on the market prices to be insignificant. We search for a system that maximizes expected gains without any kind of human input, besides the choice of parameters and, of course, the programming.

To do so, current implementations use various strategies and programming paradigms to work under certain boundary conditions. These agents employ different algorithms and need different kinds of inputs to work in their respective environment, such as to predict prices in a stock market (Jeong and Kim 2019). In essence, an agent needs to know what to trade and in what kind of space they are trading, such as the ask-and-bid spread in the underlying market. As we are dealing with time series, the agent also needs some kind of time outline so that they understand trends and outlooks in the market. The input information, as presented via a time-based input, can vary vastly for every market (Jeong and Kim 2019).

We focus on a derivatives market for Contracts for Differences. To investigate the effect of market information given to the agent, this paper compares varying degrees of input length on one implementation of a suitable market trading agent.

We want to find out the effect that different amounts of information can have on the agent, and whether it can positively or negatively influence the strategy taken by the agent. Either effect should be shown by the agent's ability to gain profit on a given market, with a specific time frame as an observation space for the strategy.

One could assume that if the agent is able to access longer time periods of market information, it would be able to draw better conclusions on the trends in the market. Interpreting our results, we can state that this is not the case. To see if the Reinforcement Learning agent performs well, we will compare it with a hold-buy agent and a random action agent. The implementation is based on modeling the problem as a Partially Observable Markov Decision Process (POMDP) optimized with a Q-Learning approach. Thus, instead of forecasting the market, the agent will have a specific action space from which it can sample trading actions to gain more profit on a given market. These actions include buying and short-selling the contracts, as well as waiting. Further information on the setup and the agent model can be found in Section 3. Overall, this paper starts with a short introduction into the general research done in this area and how other agents are getting the market information, found in Section 2. Section 3 will give a detailed run-down of the research done, the evaluation setup for the agent, and how the agent works with the Q-learning implementation. Afterwards, in Section 4, we present a short overview on the insights gained and discuss possibilities on how to progress with the research.

## 2. Literature Review

Research in the domain of automatic trading of financial assets is a wide field with various papers and implementations, merging computer science and economics (Chakole et al. 2021; Golub et al. 2018; Jeong and Kim 2019; Kearns and Ortiz 2003; Zengeler and Handmann 2020). On the side of computer science, the advent of artificial intelligence brought with it a new focus on agents that could deal in the stock market on their own and make profit. This leads to different directions of research, some of which apply Reinforcement Learning (Chakole et al. 2021; Zengeler and Handmann 2020). Yet, the effects of observation time remain, often as an open question that needs more research.

While there is a lot of new research being done on reinforcement learning in the context of financial markets, especially on research questions like profitability, as stated by (Meng and Khushi 2019), the overall research in the field is in a rather early stage and needs more scientific attention. A lot of studies focus on general algorithms and how they can be used, and only a few focus on the inputs and how a different observation period can maybe have an influence on the actions taken. It is still possible to find details on how the input is modeled in other papers, and a few different approaches can be taken away that have a unique way of dealing with time-based inputs.

The research done by (Golub et al. 2018) dismantles time itself in favor of a general trend. Here, the agents reason on periods of growth and decay in the form of trends. More granular, minute changes in the stock market are found to lose in favor of much broader trends, which represents the stock market in a wider sense.

Another smaller Reinforcement Learning approach tracks every second of the market history with both the highest ask and lowest bid price (Zengeler and Handmann 2020). This information, available to the agent as a time series of ticks, serves as an observation space for reasoning on a policy. The observation consists of a sequence of close per-second looks at the market over a given period of time.

Other strategies include algorithms that take certain observation times into consideration and will generate different predictions and conclusions based on expert knowledge (Kearns and Ortiz 2003). These approaches do not consider self-learning agents, but agents built for a specific purpose, which act according to a basically rule-based algorithm. Nonetheless, these programs need a certain amount of temporal information about the current market. The information needed for this differ in a few ways from the information used in this paper. Some algorithms access trend indicators, while other algorithms use additional information, like market volatility and trend-based predictive strategies (Kearns and Ortiz 2003). These observation periods are no longer just based on the ask-and-bid spread of the market, but take further information into consideration. In a general survey by (Fischer 2018), different models were compared that feature a variety of methods to make use of inputs. These approaches make use of different strategies relying

on different inputs, but no further information is given on how the agents could perform with different durations of observation.

While the agent in this paper is tasked to make decisions on the current market to try and gain a profit, there are different variations of market research involving machine learning algorithms. A look into other research fields shows that there are some observations on input time. The following article (Cervelló-Royo and Guijarro 2020), tries to forecast the market based on different machine-learning algorithms, and found out that at least while forecasting a longer time-period, the forecast accuracy is higher. Another article (Weng et al. 2018) shows that even reliable one-day forecasts with a mean absolute percent error of $\leq 1.50\%$ are possible, but require different datasources and information.

## 3. Methodology

To find out how and whether the observation period can have an influence on the trading agent, we have performed training and test runs according to the methodology described in this section. Instead of general market trends, as mentioned in Section 2, our approach operates on one-second time frames. We use and improve upon the base setup from (Zengeler and Handmann 2020), in which the observations last for exactly five minutes, in the form of 300 ticks. Each tick contains the ask-and-bid prices and volume data per second, and for each asset we take trading into consideration. The agent uses these ticks as an observation in order to render the next action according to their policy. Instead of trading only on observations of a single asset, the agent can draw conclusions between a multitude of assets. For our tests, we evaluate the performance of four different assets: OIL, GOLD, EURUSD and US500. Using a broad observation space of all four recorded assets, our optimization model may capture information from correlations. To take advantage of correlations, we introduce a Convolutional Neural Network (CNN) prior to a recurrent Long Short-Term Memory (LSTM) part. Optimizing CNN kernel parameters instead of fully connected layers yields improvements in training speed and model size. We have also improved on the reward scheme of the base setup, such that every trade will yield a reward $r \in \{-0.1 \ldots 0.1\}$, representing a percentage of an increase or decrease in equity. This leads to better comparability, as we may measure success independent of concrete pricing. After reaching the limits of either 10% profit or loss, or at the end of the day, the trade is closed.

### 3.1. Data Setup

We recorded trading information over a period of time from 25 May 2020 until 27 July 2020, between 11:00 a.m. and 6:00 p.m. each business day for a total of 43 different Contracts for Difference (CfD). In the recorded market phase, we observed a general recovery from the stock exchange crash of March 2020, as showcased in Figure 1. Overall, markets were in a rather unexpected, strong up-momentum. Detailed charted market data for the assets that we have used can be found in the Appendix A. While we did record a total of 43 different contracts, only four were chosen for the agent to learn and test on, and these assets are: *OIL, GOLD, EURUSD*, and *US500*. These assets vary vastly in what kind of financial assets they represent, which is also the reason why they were picked out of the 43 total recorded assets. The *US500* relates to the Standard & Poor's 500 stock market index, which includes the top 500 stock traded companies in the United States of America, while *GOLD* and *OIL* both rely on the respective real-world commodities. The *EURUSD* asset comes from a foreign exchange market between the Euro and the US Dollar. To evaluate the agents' generalization performance, we split our data into training and test data sets, such that every third day added to the test data set instead of the training data set. The Appendix A contains figures that show all daily closing prices for every asset picked and trained on.

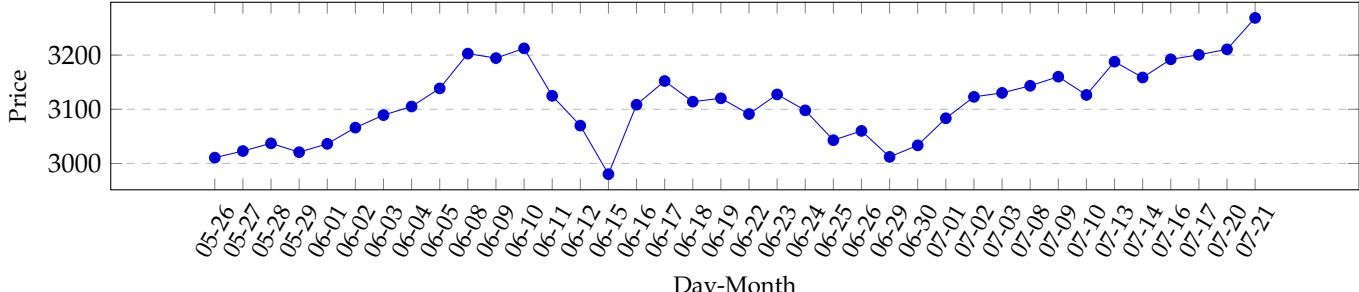

**Figure 1.** Charted information for US500 market, year 2020.

### 3.2. The Agent

The agent uses an input array of a fixed size as an observation tensor, which consists of ticks that capture a consequent time frame of prices of a number of assets. Each tick lasts one second and contains both the ask and bid prices. As elaborated, a Reinforcement Learning agent may interact with a market environment by taking actions out of an action space that will result in a financial profit or loss. The action space available to the agent consists of three different actions: to either wait, buy, or short-sell contracts. After it reaches either a 10% percent profit or loss, it receives a reinforcement signal based on the performance.

Based on the input array given to the agent, we can define a certain observation period and see how that affects the actions chosen. A longer input, such as 600 ticks, would mean that the agent has the last 10 minutes of the market as input information, while shorter input arrays mean a short observation period for the agent. With this, we can regulate how much information we give to the agent before it has to take any action. This also defines the basis of this test, as we regulate the amount of information fed to the agent and see how this will influence the actions taken and profit gained.

### 3.3. Evaluation Setup

To evaluate the observation time effects, we train Q-Learning agents for each combination of observation duration and asset. Then, we run 25 tests on the test data set to level out inconsistency. Based on this, we have systematically adjusted the observation length to warring degrees in order to use different lengths of observation time frames. Each one of 25 testing cycles starts with a certain amount of equity. We then perform 100 test trades per cycle, in each of which the agent makes a decision on their given information and waits for the market environment to yield a profit or loss. At the end of each cycle, we check whether the total equity has increased or decreased. If an agent closes a test run with a positive increase in equity, we count it as a win. As an indicator for the performance of an agent, we simply count the number of tests with financial wins or losses. We did not consider the total amount of equity won or lost, but just counted the signs of profit. This allows us to compare a wide spread of possible observation time-scales, as the total success does not rely on concrete prices and leverages. One example for this can be seen in Figure 2, where the total margin did rise a substantial amount, but fell down to being just a small amount above the starting point, and this was still counted as a win for the agent, especially considering possible applications for a high-frequency trading setting, in which even smaller periods of time can decide whether it was a win or a loss. A flowchart showing the evaluation setup can be found in Figure 3.

We have chosen the following observation periods: 10 s, 30 s, 45 s, 60 s, 5 min, 8 min, 10 min, and 12 min. With that, we capture small periods of observation, like 10 seconds, as well as rather long periods of observation time, with an input length of 12 minutes. We only change the length of observation and the input size for the input layer, as this has to change based on the given observation time. We did not change other hyper-parameters, like the learning rate or batch size. On the other hand, we also left out longer periods of time, like one hour, six hours, or even days or weeks.

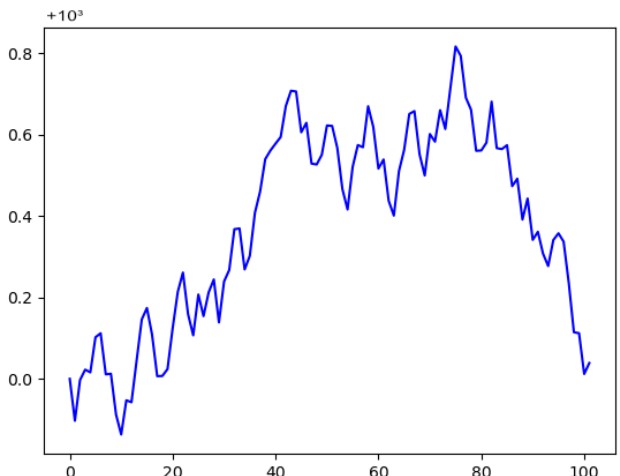

**Figure 2.** An exemplary margin chart from a test run on *EURUSD* in 45 s.

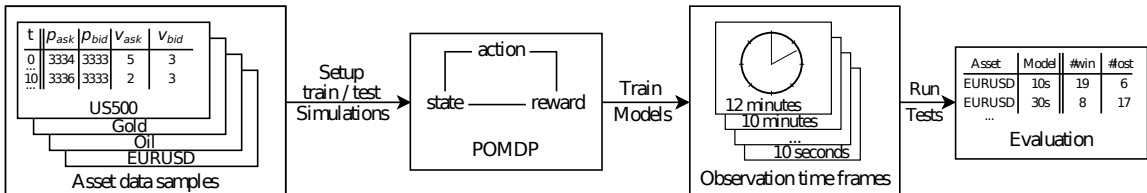

**Figure 3.** A sketch of the whole workflow setup for sampling, training and evaluation.

We perform a test for each of our four assets, with 25 test runs per asset and a total of eight different observation periods. This means that per period, we have a total of 100 test cycles to compare for the overall performance. In total, we evaluated 800 trials to find out what effect the observation period has on the agent. With 100 trades per trial, we conducted a total of 800,000 test trades. To compare the results gained by our Reinforcement Learning method with reasonable baselines, we made use of two other agents, acting in the same environment but following only a simple buy-hold or random strategy. One agent was programmed to use a buy-hold strategy, in which the first action chosen on the dataset was always a buy action. The other agent was implemented based on choosing a random action and executing this action. This was done to gain a baseline understanding and see if the main agent in question was performing on a decent level, thus achieving more wins than just, for example, taking random actions on the market.

### 3.4. Q-Learning Model

Here we use a Q-Learning approach with prioritized experience replay memory and a deep neural network according to (Zengeler and Handmann 2020), which enables agents to interact with a simulated CfD market environment. In our simulation, each action will garner a reward $r \in \{-0.1 \ldots 0.1\}$ that drives policy optimization. The agent estimates the rewards from the environment given this action, thus finding the most profitable action for a given observation.

In general, a Reinforcement Learning agent acts upon a set of observed information from an environment. As we do not know the order book, we need to provide everything that the agent needs to make a decision and gain rewards under the limitations of a Partially Observable Markov Decision Process (POMDP). In the case of automatic trading, that observation may capture information like the current bid and ask prices of a number of assets. This information, bound in a certain time frame, allows the agent to reason upon a decision on whether to buy or short-sell an asset. In the case of the agent used in this paper, they have the ability to select an action to buy ($a = 1$) or short ($a = 2$) contracts, or to wait ($a = 0$).

Figure 4 illustrates the neural network architecture used in our evaluation. We make use of a CNN and a LSTM network. The use of a CNN allows us to capture correlations between the single assets. Though, in theory, we may also capture correlations with a fully connected layer, optimizing CNN kernel parameters exploits temporal neighborhood features and reduces training time and model size. For our kernels, we do not assume any neighborhoods between the assets, but only their temporal correlations. For example, we might expect a rise in gold prices when we observe a decrease in index values, and vice versa. To optimize our weights in the model, we are using the Adam optimization algorithm (Kingma and Ba 2017). The Adam optimizer was picked for its efficiency and good handling with larger input data. Another important factor for our learning model are the hyper-parameters that were used for the training process. Our parameter for batch size was put at 30 to keep the memory cost of the longer observation-time training sessions low. The learning rate was set at $10^{-5}$, a value taken after testing multiple other values and checking the losses incurred, as seen in Figure 5. All other graphs for the 45 s time frame can be found in the Appendix C.

A closer look at the agent, hyper-parameters, and model is furthermore possible in the Appendix B with a short pseudocode excerpt and the linked GitLab repository.

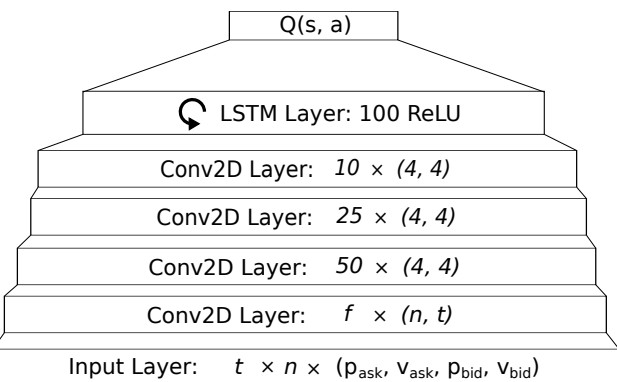

**Figure 4.** The neural network architecture used in our evaluation. For a number of observed assets $n$ over a time span $t$, the input layer consists of $(t \times n \times 4)$ input neurons. For the following convolutional layer, we choose the number of filters $f$, such that the resulting activation shape fits into the later convolutional layers. After the convolutional part, we employ a LSTM layer, consisting of 100 recurrent neurons with rectified linear activation.

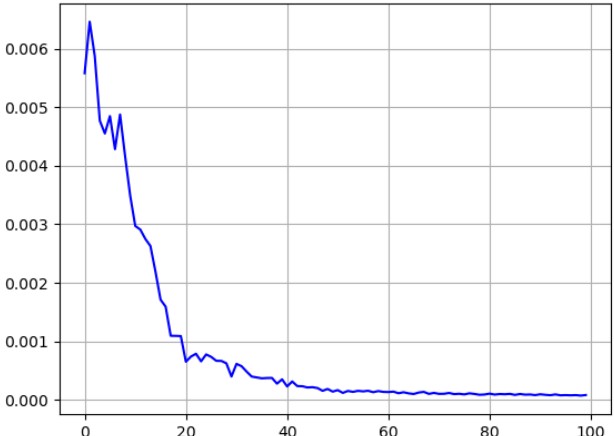

**Figure 5.** Example loss graph for one training period: *US500* 45 s.

## 4. Results

To evaluate our results, using the methods as described in Section 3, we looked at the number of wins that the agent achieved per asset. For each observation length and asset,

we ran 25 tests and counted it a win if we found an increase in total equity after a test run. This should allow us to see if there are noticeable changes between the input lengths given.

Figure 6 shows the number of trials which resulted in an overall positive profit, counted for all observation periods and all assets. As we interpret Figure 6, we do not find a general trend, but rather a distribution with different peaks, with the most notable one at 10 min where both the *US500* and *EURUSD* asset achieved profits. Considering the general performance across all assets, we find no conclusive evidence of certain periods influencing the trading outcome. Figure 7 showcases the total results per asset, summed across all time spans. This view of the results indicates that our approach works best at the *EURUSD* foreign exchange market, while the *US500* index takes second place, closely followed by the *GOLD* commodity market at the end with only a total of 15 wins sits *OIL*. One interesting fact to note is that our agent was unable to achieve major wins on the *OIL* market.

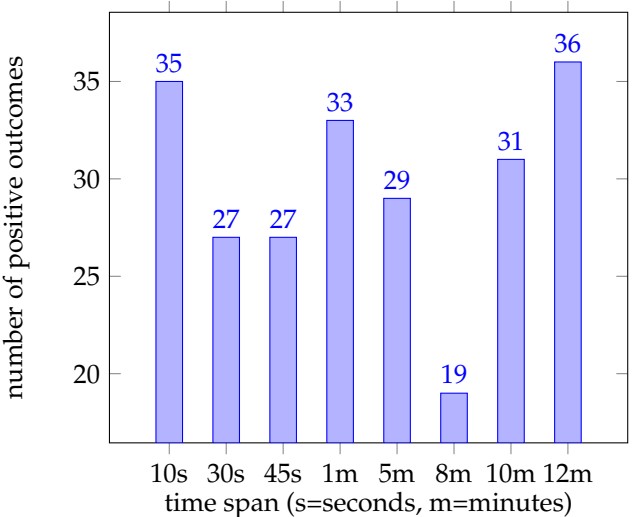

**Figure 6.** The number of positive outcomes over all tests with a certain observation period. As shown in the chart, one can find the highest distribution of wins at the 12 min mark, closely followed by 10 s, while by far, the lowest distribution is found at the 8 min point.

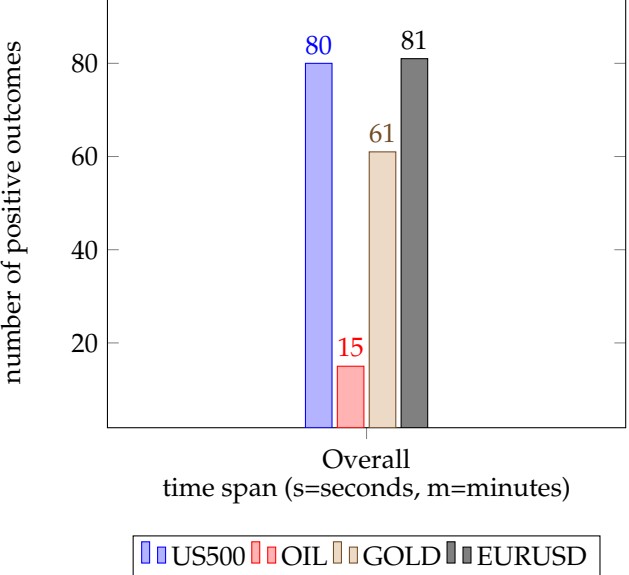

**Figure 7.** The total number of positive outcomes per asset, added across all time spans. We find a hardship in trading on the *OIL* market, but a successful application in the *EURUSD* and *US500* foreign exchange.

To elaborate, we take a closer look at the individual assets and the amount of wins per temporal resolution. As we can see in Table 1, on a per-time base, *OIL* did not perform well for any time length given. On the other hand, we can see that *EURUSD* and *US500* performed rather well. *US500* performed rather well at both the 45 second and 1 minute input lengths, closely followed by the performance at the 10 second length. The *EURUSD* asset had the most outstanding success, with a total of 19, leading us to the assumption that this concrete approach and model suits fine for foreign exchange trading.

As already stated in Section 3, to check if the agent and approach used in this paper has any basis, we conducted tests with a buy-and-hold strategy, as well as a random strategy. To prelude, these agents did not perform as well as our Reinforcement Learning agent, as shown in Table 2. The hold-and-buy agent would probably have had better performance in an even longer data set and without the 10% and daily closing constraints for determining if an agent made a profit. The random strategy achieved a solid performance on the *US500* market, followed by both the *GOLD* and *EURUSD* markets. Still, it was not able to gain any significant amount of wins on the *OIL*. These statistics, as seen in Table 2 for the other agents and in Table 1 for the Reinforcement Learning agent, show us that overall, our approach took more profound actions and in turn was able to make more consistent profits on most markets. As it stands, only the *OIL* market can be regarded as a complete loss considering no noticeable wins were achieved, regardless of the agent used.

**Table 1.** A closer look at the details. Each cell contains the total number of positive outcomes (maximum 25), for each combination of asset and observation time frames. Highlighted in bold are assets that achieved a substantial amount of wins in their observation time.

| Total Number of Wins Per Asset | | | | |
|---|---|---|---|---|
| **Time** | **US500** | **OIL** | **GOLD** | **EURUSD** |
| 10 s | 10 | 1 | 5 | **19** |
| 30 s | 7 | 5 | 7 | 8 |
| 45 s | **13** | 0 | 6 | 8 |
| 1 m | **15** | 3 | 11 | 4 |
| 5 m | 9 | 1 | 6 | **13** |
| 8 m | 6 | 1 | 7 | 5 |
| 10 m | 11 | 1 | **12** | 7 |
| 12 m | 9 | 3 | 7 | **17** |

**Table 2.** Other agents and performance on the given dataset (maximum 25).

| Total Number of Wins Per Asset | | | | |
|---|---|---|---|---|
| **Agent** | **US500** | **OIL** | **GOLD** | **EURUSD** |
| Random | 7 | 0 | 7 | 10 |
| Buy-Hold | 0 | 0 | 1 | 23 |

## 5. Conclusions and Outlook

To conclude, we state that we did not find an overall optimal observation duration for Reinforcement Learning approaches on Contracts for Difference. Of course, we can only reason on the simulation used in this paper and the given observation period of 10 seconds to 12 minutes. We could not find an overall trend in the amount of wins achieved that would depend only on the observation time. Using different observation periods, none of them indicate a general improvement in having a longer or shorter period of information input. Rather, there seem to exist some observation sequence lengths that work better than others. These performances include the 10 second input length on *EURUSD* performance, as well as the 45 second and 1 minute performance on the *US500* market. While certain periods of time had better performance on some assets, no general trend can be observed.

This also means that we did not find a regularity between different timed observation inputs. Interestingly enough, no agent performed well on the *OIL* market; instead, each investigated optimization model made constant losses, with only a few wins. This could be based on the dataset used, but needs further exploration.

Other open questions remain, as we did not explore the variation of hyper-parameters that may have altered the agents' learning behavior, as explained in Section 3. Considering the effects of changing the observation time span, the kind of influence that daily or even weekly periods could encompass remain unknown for our setting. A larger time span may level out small but frequent spikes in the market, but leave longer overarching spikes. As already discussed in Section 2, the article (Cervelló-Royo and Guijarro 2020) shows that, at least for forecasting a longer time period yields better results, it could be possible that these results can be transferred into this research topic, but at least with our findings we cannot make conclusions in the researched time span of 10 s to 12 min. No trend was found that longer observation periods yield better results. This could also be based on the researched time spans being rather close, where maybe time spans involving multiple hours or even days show different results. One interesting topic of future research could be the combination of various machine learning algorithms. One example would be to train an algorithm on forecasting, and afterwards feed the decision-making agent with the predicted information, together with the currently observed market observation. We would also like to explore the application of our system in high-frequency trading, which takes place in the range of microseconds. Unfortunately, we lack the necessary data, as the data we have used was a a one-second time scale per tick, but this is still not enough to consider this research as high-frequency trading. Furthermore, we did not investigate any other forms of input that may benefit shorter or longer time periods, which we leave for exploration in future papers, especially considering an agent that uses input information given as general trends, as seen in (Golub et al. 2018).

Thinking even further about this idea suggests transferring the observations into a latent space, for example with a tailored Variational Autoencoder (VAE). With that kind of input form, we may try to predict the next latent vector, thus making a model of the market without any reinforcement signals. We may then use a low-parameter controller network to transfer world model knowledge into a concrete policy for arbitrary trading systems. Such a Transfer Learning approach comes with the benefit of temporal invariance, explainable policies using world model knowledge, and fast transfer to new tasks. Going forward, it would be interesting if this test could be done with different input models, like the ones already explored in Section 2. While the dataset would most likely need to be transformed to make it fit, like turning it into general trends, this should still allow us to draw a more detailed conclusion on the effects of input length.

**Author Contributions:** Conceptualization, M.W.; methodology, N.Z.; writing—original draft preparation, M.W.; writing—review and editing, N.Z.; supervision, U.H.; project administration, U.H. All authors have read and agreed to the published version of the manuscript.

**Funding:** This research received no external funding.

**Institutional Review Board Statement:** Not applicable.

**Informed Consent Statement:** Not applicable.

**Data Availability Statement:** The recorded market data together with the source code and hyperparameters can be found at https://gitlab.hs-ruhrwest.de/nico.zengeler/cfd_src.

**Conflicts of Interest:** The authors declare no conflict of interest.

**Sample Availability:** All source code and data samples are available from the authors or at our GitLab repository at https://gitlab.hs-ruhrwest.de/nico.zengeler/cfd_src.

## Abbreviations

The following abbreviations are used in this manuscript:

CfD     Contracts for Difference
CNN    Convolutional Neural Network
LSTM   Long Short-Term Memory
VAE    Variational Auto-Encoder
MDN    Mixture Density Network

## Appendix A. Graphs

Charted information for US500, year 2020

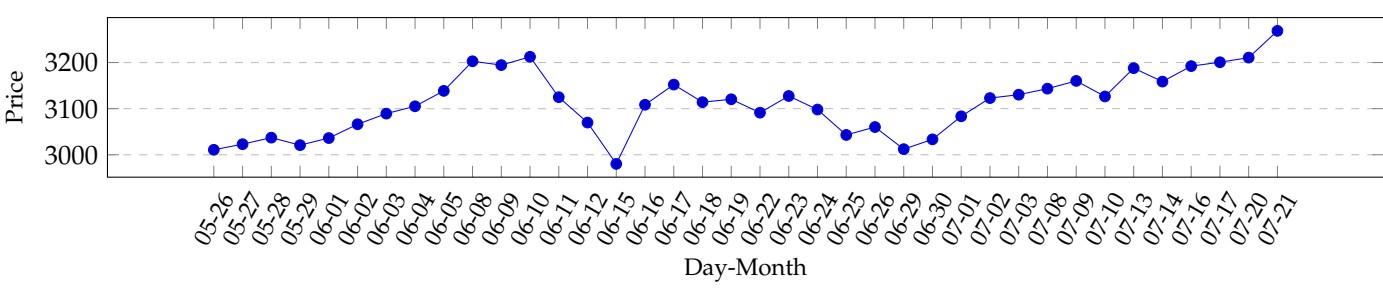

Charted information for Gold, year 2020

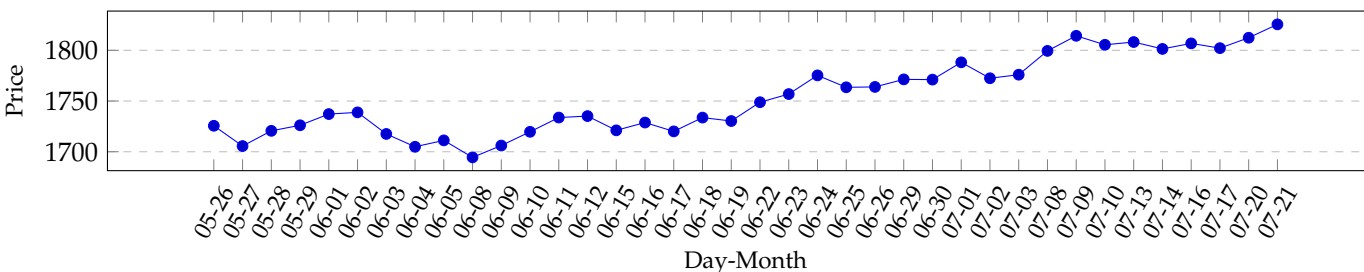

Charted information for EURUSD, year 2020

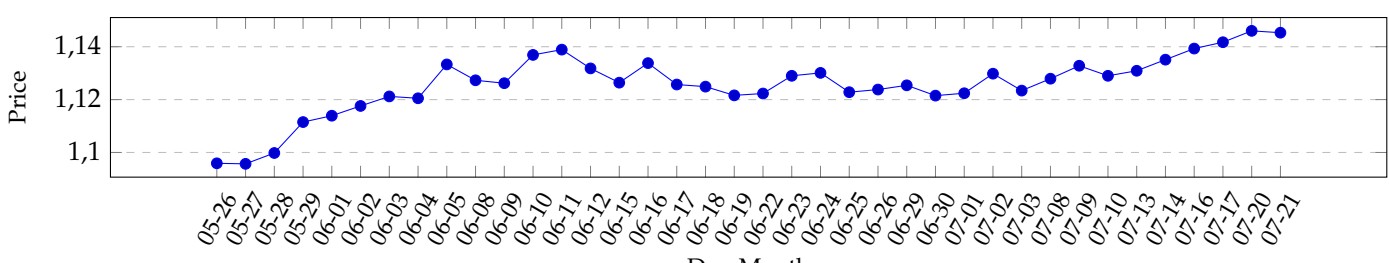

Charted information for OIL, year 2020

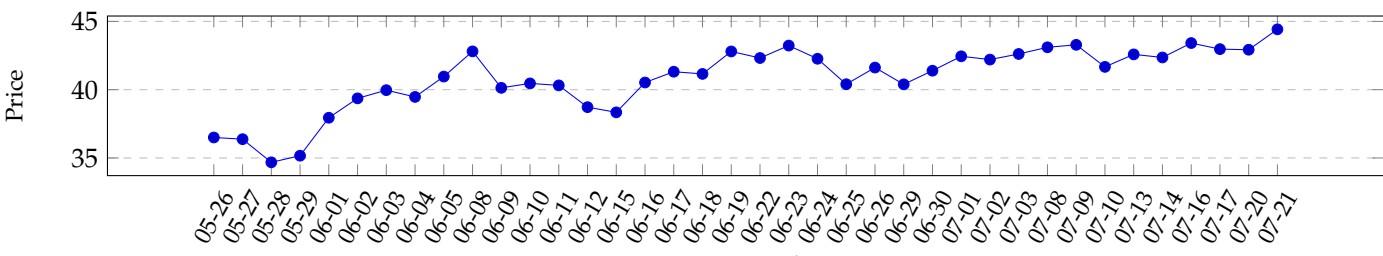

### Appendix B. Pseudocode

---

**Algorithm A1** Training in Market Simulation

---

1: **procedure** TRAINING

2:     *Historic trade data X*

3:     *Prioritized experience replay memory M*

4:     *Input sequence length L* **adapted for the observation length**

5:     *Policy $\pi(s|\Theta)$*

6:     **while** *learning step < number of steps* **do** iterate of random points of time

7:         *$t \leftarrow$ random point of time in market history*

8:         **while** $t \leq len(X) - L$ **do** choose action based on point in time and observation
length

9:             *$state_1 \leftarrow X[t : t + L] - mean(X[t : t + L])$*

10:             *$action \leftarrow \pi(state_1)$*

11:             *$(state_2, reward, t, terminal) \leftarrow marketLogic(action)$*

12:             *$appendToMemory(M, (state_1, state_2, action, reward, terminal))$*

13:             **if** $|M| \geq batchsize$ **then** increment learning step and update the model

14:                 *$batch \leftarrow sample(M)$*

15:                 *$losses \leftarrow Q - Learning(batch)$*

16:                 *$updatePriority(M, batch)$*

17:                 *learning step $\leftarrow$ learning step + 1*

---

### Appendix C. Randomly Chosen Loss Graphs

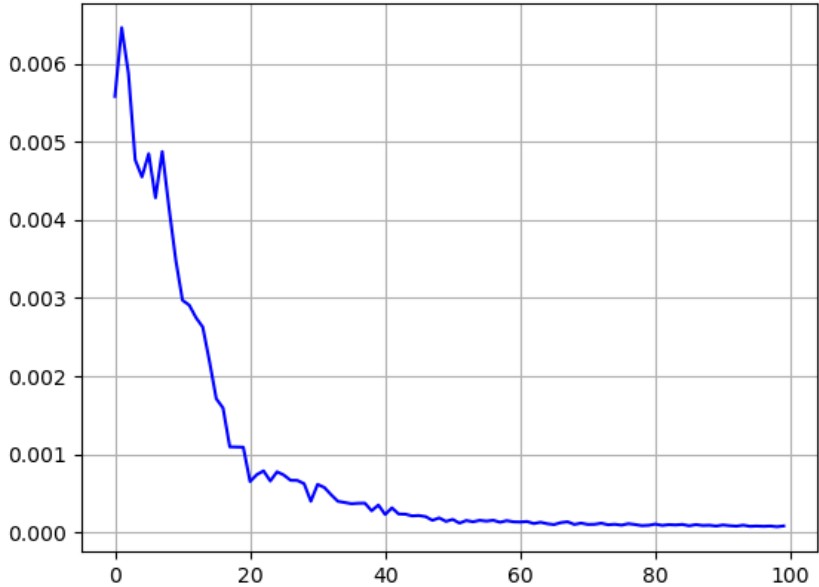

**Figure A1.** *US500 Losses*, 45 s.

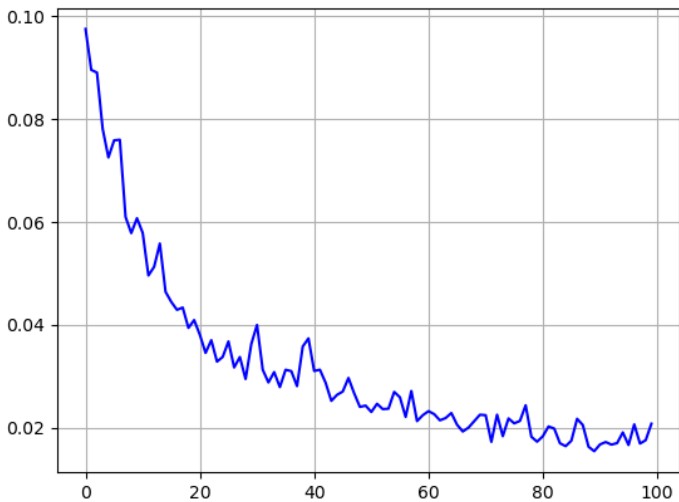

**Figure A2.** *GOLD Losses*, 45 s.

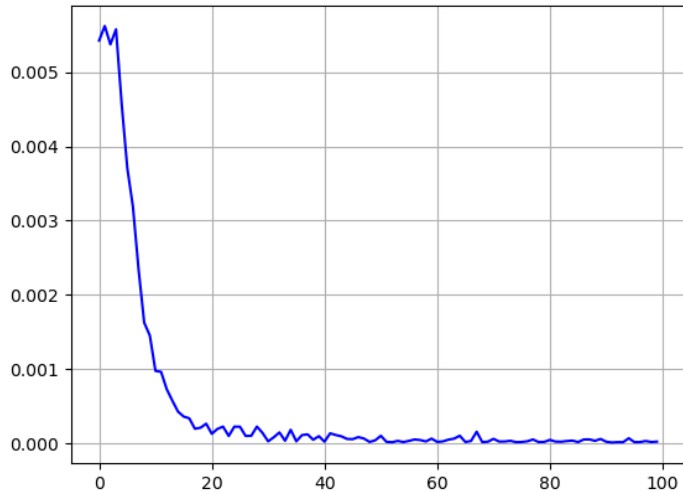

**Figure A3.** *OIL Losses*, 45 s.

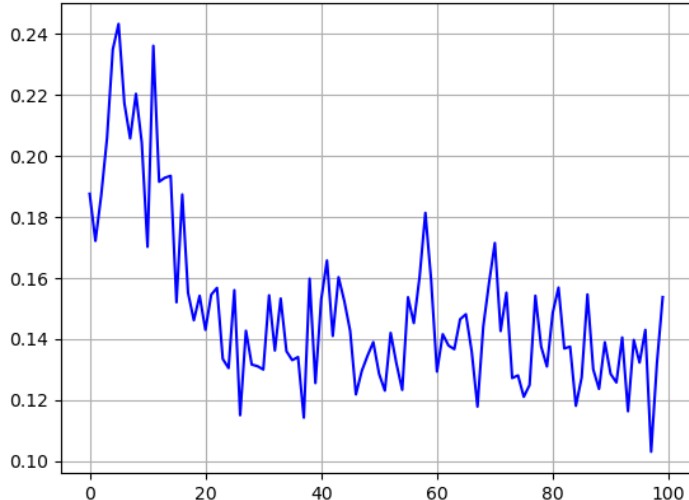

**Figure A4.** *EURUSD Losses*, 45 s.

## Appendix D. Manually Chosen Margin Graphs

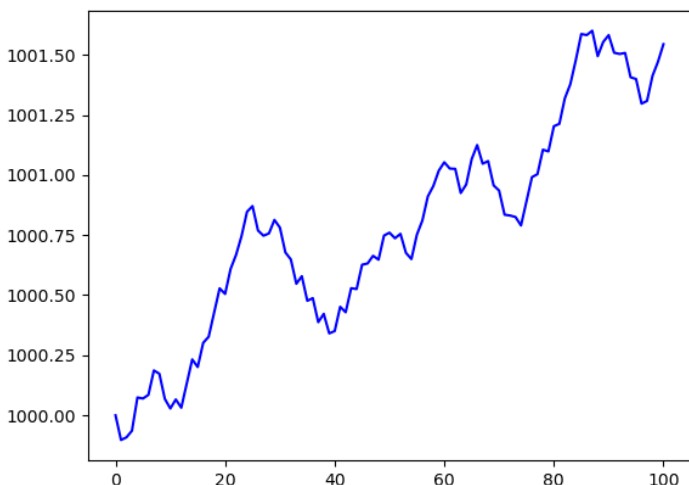

**Figure A5.** *EURUSD Margin*, 10 s.

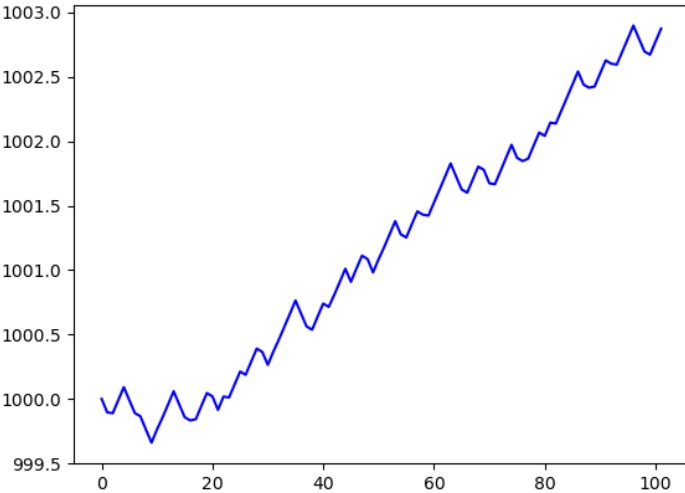

**Figure A6.** *US500 Margin*, 45 s.

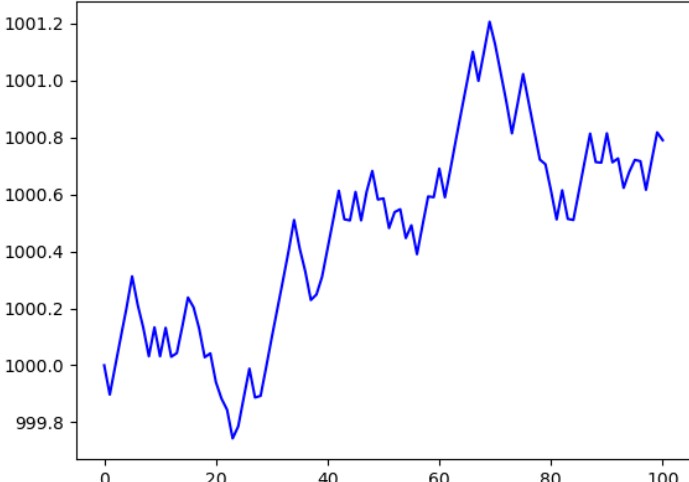

**Figure A7.** *US500 Margin*, 60 s.

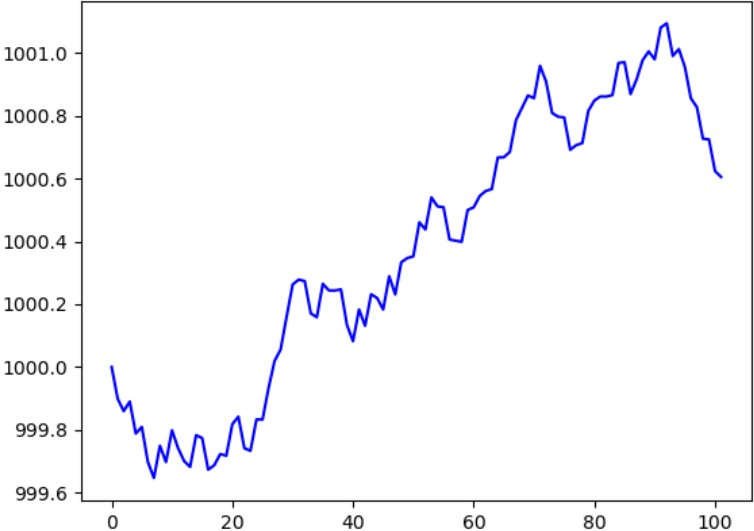

**Figure A8.** *EURUSD Margin*, 5 m.

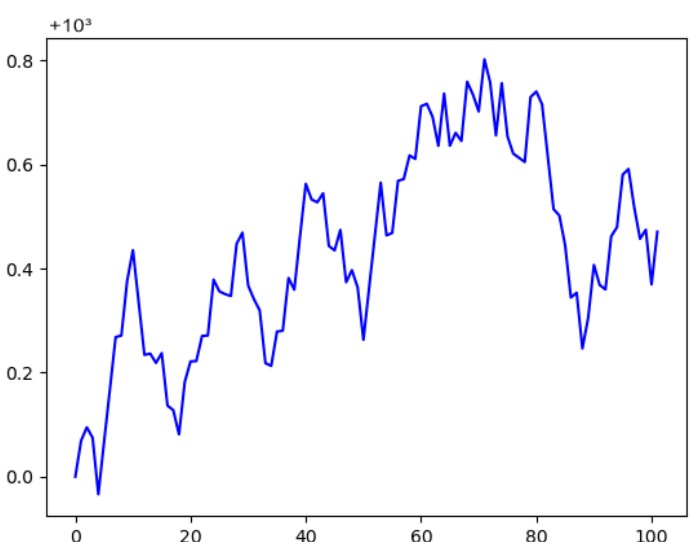

**Figure A9.** *GOLD Margin*, 10 m.

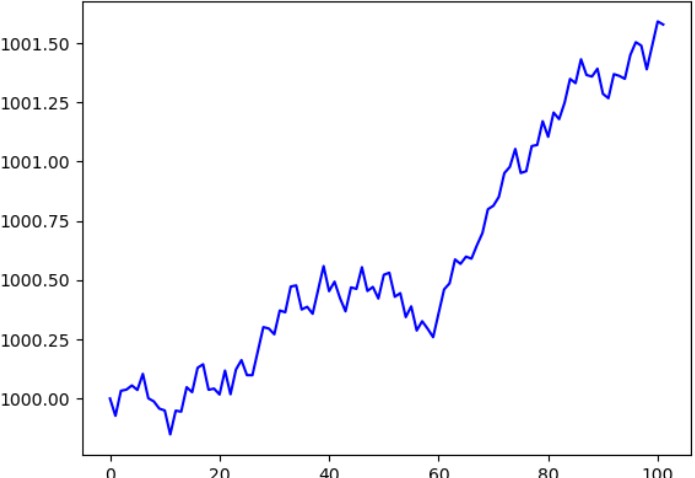

**Figure A10.** *EURUSD Margin*, 12 m.

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
