# Peer review of "Observation Time Effects in Reinforcement Learning on Contracts for Difference"

_jrfm, doi:10.3390/jrfm14020054_

Round 1

Reviewer 1 Report

Its a very good paper, that can be accepted in current form.

Author Response

Thank you very much!

Reviewer 2 Report

The literature review section should be improved. There is a large list of papers dealing with automatic trading and expert systems. The papers highlighted by authors constitute a very small sample of the research devoted to this topic. I would suggest to include this reference and some of the papers cited in here: https://doi.org/10.46503/NLUF8557.

Quality of figure 1 should be improved.

The methodological part should include a picture that summarizes the whole process. Authors show some figure to illustrate the neural network architecture, but the paper lacks of a picture to describe the whole process to perform the research.

The performance of the trading rules should be better explained. We should know the return and risk of the strategy.

Author Response

Thank you for your feedback. We have included and discussed the possible implications of this reference and referenced one of the references references. We have improved the quality of figure 1 by removing vast space between data points. We have added a figure that depicts the whole workflow in our methodological section. We have chosen exemplary margin graphs to illustrate the performance of the trading rules. As mentioned in the text, a reader may find source code to reproduce in our GitLab repository.

Reviewer 3 Report

The paper is well written but needs to have a simulation study.

Author Response

With the new figures and explanations, we hope to have pointed out the setup of our simulation
which we have studied. As we have splitted our dataset into a training and testing set for the random sampling of observations in the simulated environment, we would have observed bias effects like overfitting.

Round 2

Reviewer 2 Report

Authors have addressed all my concerns

This manuscript is a resubmission of an earlier submission. The following is a list of the peer review reports and author responses from that submission.

Round 1

Reviewer 1 Report

The study suffers major flaws in its design and analysis that the conclusions can not be supported by the presented data.

Reviewer 2 Report

The introduction section serves both as an introduction to the topic and as a short literature review. This is not bad in essence, but I miss some paragraph devoted to the explanation of the gap research, the aim of the paper and a short sentence related to the main conclusions raised by the research. This would help the reader to understand the proposal in just a few lines, and then decide whether or not the reader is interested in the topic addressed by the authors. Furthermore, a deeper introduction regarding IA methods applied on stock markets should be included (https://doi.org/10.46503/NLUF8557).

Line 89. The sentence "for a total of 43 different Contracts for Difference (CfD)" is a bit confusing. Authors state that the research covers four different assets: OIL, GOLD, EURUSD and US500. In that line the reader can be confused regarding the number of assets traded in the research. Are they just 4 assets of 43 different derivatives from these assets? Please state this issue in a clear way.

Line 147. I miss a point symbol in the middle of "CFD market environment In our simulation".

Line 159. Authors state: "The use of a CNN allows us to capture correlations between the single assets". Is there any other option of neural network capable of capturing correlations between assets, of CNN is the only one with this capability? A more detailed explanation on the neural network chosen is appreciated, please focus on the pros and cons of this network compared with other approaches.

Other issues being important, my main concern is related with the results Section. All figures and tables summarize the output of the trading strategy. The number of trades is too low to infer any conclusion regarding the actual profitability of the strategy. We cannot consider a maximum of 10 trades as positive outcomes (Figure 4). We need many more trades to get statistically significant results. Authors have not performed whether the strategy has data snooping issues. They should include a longer horizon, so that a larger dataset of trades is saved and results are analyzed from a statistically point of view.

Reviewer 3 Report

  • It's not clear what authors would like to deliver in this study. The title doesn't convey the overall picture and the significance of the study.
  • The introduction section doesn't describe the problem well. Also, it's unclear what the contribution of this study is.
  • There is no connection between the literature review and the methodology.
  • The methodology needs to be better described and indicates how it can be used to find the answers.
  • The results can be better explained. No implications from the results are discussed.